# Detrimental Health Behaviour Changes among Females Living in Rural Areas during the COVID-19 Pandemic

**DOI:** 10.3390/ijerph18020722

**Published:** 2021-01-15

**Authors:** Kristen M. Glenister, Kaye Ervin, Tegan Podubinski

**Affiliations:** 1Department of Rural Health, University of Melbourne, Docker Street, Wangaratta, VIC 3677, Australia; tegan.podubinski@unimelb.edu.au; 2Department of Rural Health, University of Melbourne, Broadway Street, Cobram, VIC 3644, Australia; keervin@unimelb.edu.au

**Keywords:** rural, COVID-19, pandemic, health behaviours, coping, exercise, alcohol, smoking, gender-responsive

## Abstract

Women are predicted to be disproportionately impacted by the COVID-19 pandemic due to increased carer responsibilities, loss of income, worry about the virus and a predominantly female healthcare workforce. Whilst there is emerging evidence that negative mental health impacts associated with the COVID-19 pandemic may be more pronounced for women than men, less attention has focussed on changes to health behaviours and health seeking experienced by women. Similarly, the impact of the pandemic in rural areas has not been investigated in detail. Our research questions were ‘*have females residing in rural areas experienced changes in alcohol consumption, unhealthy food consumption, smoking, exercise or health seeking during the COVID-19 pandemic?*’ and ‘*are there differences in health behaviour changes between rural females living with or without children?*’. Net increases (scale of 0–1) in consumption of unhealthy food (95% CI 0.05, 0.22) and alcohol (95% CI 0.12, 0.29) were observed. Net decreases (scale of −1 to 0) in visits to the doctor (95% CI −0.23, −0.35) and other health professionals (95% CI −0.40, −0.54) were observed. Compared with females living without children, females who lived with children were significantly associated with increased alcohol consumption (OR 2.4 (95% CI 1.4, 4.1), decreased visits to the doctor (OR 1.9 (95% CI 1.1, 3.2) and decreased visits to other health professionals (OR 1.9 (95% CI 1.1, 3.3). Results suggest that public health approaches may be required to support females residing in rural areas to optimise their health behaviours during the pandemic, particularly for those living with children. Policies must be gender responsive to counteract worsening health and social inequities both during and after the pandemic.

## 1. Introduction

Pandemics and public health responses to control spread of infection can disrupt everyday life and have consequences for health and well-being. Pandemics, including COVID-19, have the capacity to worsen pre-existing inequities in health and social outcomes among vulnerable groups of people [1]. It has been widely predicted that women will experience greater impact from the current pandemic than men, due to increased likelihood of loss of employment or income, increased carer responsibilities due to closures of schools and childcare facilities and the predominantly female health and social workforce on the front line [2]. These changes may contribute to increased stress and changes to health behaviours. This study will focus on the health behaviour impacts for females residing in rural areas, with or without children, for whom access to health services and health promotion activities are likely to be constrained compared with their metropolitan counterparts.

### 1.1. Female Mental Health and Health Behaviours

Increased psychological distress associated with the pandemic and widespread restrictions have been reported internationally and appear to be impacting females more than males. Internationally, studies from China [3], United States [4], United Kingdom [5], Canada [6] and Australia [7] have each reported increased symptoms of poor mental health among females compared to males, and for some studies these changes were particularly pronounced for female carers [4] and females living with children [6]. Psychological distress can lead to poor health behaviours being used as maladaptive coping strategies. In a large Australian study increased psychological distress during the pandemic was associated with decreased exercise, poor sleep, smoking and increased alcohol consumption [8]. Increased alcohol consumption associated with stress has been reported to be more common among females than males [9]. A study of 1500 randomly selected participants in the Netherlands reported that unhealthy snacking was associated with coping with negative emotions, particularly among young people and women [10]. In contrast, recent surveys in Australia by a health promotion organisation and a metropolitan children’s hospital have reported an increase in home-cooked meals [11,12]. Similar findings have been reported in Spain with increased consumption of fruits and vegetables observed, potentially due to increased meal preparation at home [13]. Females often make key dietary choices for their families [14] and act as role models for their children regarding healthy behaviours [15]. For this reason, it is important to understand the impact of the pandemic in association to changes in psychological distress and health behaviour patterns among females and those caring for children, to develop targeted strategies.

### 1.2. Health Seeking Behaviours

There is evidence that the COVID-19 pandemic has been associated with changes to people’s health seeking behaviours, prompting concerns of delayed care or late presentations for important health conditions. A large study in the United States reported that 41% of respondents had avoided or delayed medical care during the pandemic, and this was particularly evident among females and adults with carer responsibilities for other adults [16]. A recent online survey of over 700 people in Australia reported that 32% of respondents had delayed or avoided a general practitioner appointment and that the predominant reason provided was that people were worried about contracting the virus during appointments [17]. Anecdotally, emergency department presentations have declined by 10–30% in Australia during the pandemic, though the reasons for this are likely multifactorial [18]. The pandemic has had an impact on the provision of antenatal and postnatal care, as well as sexual and reproductive health services. However, as pointed out by Gribble and colleagues, we need to learn from previous infectious disease outbreaks and ensure that policies put in place to manage COVID-19 do not inadvertently harm maternal and infant outcomes [19]. 

### 1.3. Females in Rural Areas

The impact of COVID-19 on the health and well-being of people residing in rural areas has yet to be explored in detail. However, it is important to understand the impact of the pandemic and restrictions to everyday life in rural areas where there may already be less access to health services and higher rates of poor health behaviours compared with metropolitan areas [20]. In addition, rural areas of Australia have borne the brunt of recent drought, floods and bushfire [21]. There is a perception that rural communities tend to have higher rates of social capital than metropolitan communities, which may contribute to community resilience during challenging times [22]. Evidence of higher levels of certain social capital elements including social cohesion, networking and civic participation was reported for rural areas compared to metropolitan areas in South Australia [23]. However, social support and networks that are considered protective against symptoms of anxiety, depression and stress have been disrupted during the pandemic [24]. Social isolation may not only contribute to negative mood but can also increase the risk of family and domestic violence [25], of which women are at higher risk than men [26].

To date, the majority of research into the impact of restrictions on everyday life during the COVID-19 pandemic has focused on the psychological impacts, and limited research has focussed on health behaviour changes (see a recent editorial [27]), particularly healthy eating. Health behaviour changes among potentially vulnerable groups have likewise received little attention to date. As highlighted in a review by Park and Iacocca, there are complex interrelationships between stress, coping mechanisms and health behaviours, and these are influenced by contextual (such as the nature of the stressor and the social environment) and personal (for example, gender) factors [28]. The research questions were ‘*have females residing in rural areas experienced changes in alcohol consumption, unhealthy food consumption, smoking, exercise or health seeking during the COVID-19 pandemic?*’ and ‘*are there reported differences in health behaviour changes between rural females living with or without children?*’

## 2. Materials and Methods

An electronic survey link was distributed to the authors’ personal and professional networks via email and social media. The authors’ networks were predominantly located in rural areas, typically in the fields of healthcare and education. Recipients of the link were invited to distribute the link within their own networks, reflecting a snowball recruitment strategy [29]. Response rate was unable to be estimated.

### 2.1. Survey Tool

An online survey was developed using the REDCap survey platform hosted at the University of Melbourne [30]. REDCap (Research Electronic Data Capture, Vanderbilt University, United States) is a secure, web-based application designed to support data capture for research studies, providing (1) an intuitive interface for validated data entry, (2) audit trails for tracking data manipulation and export procedures, (3) automated export procedures for seamless data downloads to common statistical packages and (4) procedures for importing data from external sources [30]. The survey was designed to assess changes in health behaviours and health seeking behaviours among employees or volunteers in Australia, with responses related to health behaviour change adapted from the recoding used by Stanton et al. [8]. Demographic information was collected on respondents’ age, sex, postcode of residence, employment status (full-time, part-time, casual employment, volunteer), changes to employment due to the pandemic (loss of job, working from home, reduction in pay or hours), living arrangements (living alone, with partner, children, other family, friends or flat mates). Respondents were asked ‘*Do you live with a long-term physical health condition?*’ and ‘*Do you live with a long-term mental health condition?*’. For further detail see Appendix A. The Depression Anxiety Stress Scale (DASS-21) was used to assess symptoms of stress, anxiety and depression among participants [31]. Validity of the DASS-21 tool has been demonstrated in Australian clinical and community settings [32]. DASS-21 total scores and scores for the stress, anxiety and depression subscales were calculated and categorized as normal to mild or moderate to extremely severe [31].

### 2.2. Eligibility Criteria

People were eligible to participate if they resided in Australia, were aged 18 years or older and had volunteered or had been employed at any time during the period December 2019 to July 2020. The survey was accessible from 29 May 2020 to 8 July 2020. In late May, the first Australian COVID-19 infection curve had flattened and some lockdown measures (state border closures, restrictions on the numbers of people who could gather in homes and public venues, school children learning remotely, people working from home where possible) were starting to ease.

### 2.3. Ethics Approval and Consent

The study was approved by the University of Melbourne’s COVID-19 Central Human Ethics Committee (2056921.1). Respondents were unable to enter the survey unless they provided electronic informed consent. Respondents could choose to enter a prize draw for one $100 gift-card and/or register their interest in a follow-up survey, and for these purposes an email address was required. If respondents declined to enter the prize draw or register their interest in a follow-up survey, their responses remained anonymous. 

### 2.4. Data Recoding

Responses received after the 8 July were removed from the dataset. Respondents’ postcodes of residence were recoded into the Modified Monash Model categories of 1–7 and subsequently divided into metropolitan (MMM1) and rural (inclusive of rural, regional and remote) (MMM2-7) [33]. Respondents who indicated that they lived with pre-school aged children and/or school aged children were grouped as ‘living with children’. There were 597 valid responses received, of which 339 represented females residing in rural locations. 

## 3. Analysis

Categorical variables are presented as number and percentages, while continuous variables are presented as mean, standard deviation (SD) or 95% confidence intervals (95% CI). Comparisons were made between rural females living with or without children using chi square analysis for categorical variables and independent samples t-tests for continuous variables. Net change in behaviours were analysed as 95% confidence intervals after coding decreased behaviours as −1, unchanged as 0 and increased as +1. A series of separate logistic regression analyses were conducted to assess associations between changes in health behaviours (dependent variables) and the independent variables of symptoms of depression, anxiety or stress (DASS-21 sub-scales), living with or without children, age and presence of self-reported chronic disease. The dependent variables were selected as they had shown significant differences between females living with or without children in unadjusted tests, namely increased alcohol consumption and decreased social interaction, personal care activities and visits to doctors or other health professionals. DASS-21 sub-scale scores were compared to normative values for Australian females [31] using an online t-test calculator.

## 4. Results

The results of the sub-set of rural females are reported. There were 339 valid responses received from females residing in rural areas between 28 May and 8 July 2020. The demographic characteristics and DASS-21 scores of this subset are reported in Table 1. None of the respondents reported being diagnosed with COVID-19. Forty-one percent of respondents lived with children (12.7% with pre-school aged children 32.4% with school aged children). Almost one quarter (24.2%) self-reported a pre-existing mental health condition. A majority of respondents returned normal to mild DASS scores, as per Table 1.

Further analysis revealed that the average age of females living in rural areas without children was significantly older than females living in rural areas with children (46.2 ± 14.9 vs. 42.4 ± 8.0, *p* = 0.003). There was no significant difference in the percentage of self-reported chronic disease or mental health conditions among rural females living with or without children (20.1% vs. 20.7%, *p* = 0.996 and 21.4% vs. 25.3%, *p* = 0.402 respectively). The DASS-21 scores for stress, anxiety and depression were not significantly different between rural females living with or without children (6.3 ± 4.5 vs. 5.6 ± 4.5, *p* = 0.136, 2.7 ± 3.3 vs. 2.2 ± 3.2, *p* = 0.151, 4.1 ± 4.4 vs. 3.8 ± 4.5, *p* = 0.577 respectively), as per Table A2. 

Net changes in behaviours are represented as 95% confidence intervals in Figure 1. Net decreases were observed for social interaction, personal care activities, going to the doctor and going to other health professionals. Net increases were observed for consumption of unhealthy food, coffee, alcohol and tobacco (among respondents who smoked). No net change was observed for exercise.

Respondents living with children were significantly more likely to report increased alcohol consumption than respondents living without children (50.9% vs. 30.4%, *p* = 0.001). Respondents living with children were significantly more likely to report decreased social interaction and personal care activities than respondents living without children (95.5% vs. 89.1%, *p* = 0.037 and 58.3% vs. 39.5%, *p* = 0.001 respectively). Respondents living with children were significantly more likely to report decreased visits to the doctor and other health professionals than respondents living without children (43.8% vs. 28.1%, *p* = 0.005 and 60.2% vs. 45.7%, *p* = 0.017 respectively), as per Table 2.

Health behaviour changes which were significantly different between rural females living with or without children in unadjusted tests (increased alcohol consumption, decreased social interaction, personal care activities, visits to doctor and other health professionals) were further analysed using logistic regression. Significant associations between independent variables (namely living with children) and detrimental health behaviour change (dependent variables) are outlined in Table 3. Increased alcohol consumption (OR 2.373 (95% CI 1.359, 4.145)), decreased social interaction (OR 2.883 (95% CI 1.035, 8.033)), decreased personal care activities (OR 2.342 (95% CI 1.384, 3.963)), decreased visits to the doctor (OR 1.850 (95% CI 1.076, 3.183)) and decreased visits to other health professionals (OR 1.932 (95% CI 1.118, 3.337)) were significantly associated with respondents living with children compared to respondents not living with children after controlling for differences in age, symptoms of stress, anxiety, depression and presence of chronic disease, as per Table 3. 

## 5. Discussion

The COVID-19 pandemic and restrictions associated with the public health response appear to be associated with changes to several health behaviours among this sample of females residing in rural areas of Australia. Respondents living with children were approximately twice as likely to report increased alcohol consumption, decreased visits to the doctor or other health professional and decreased social interaction or personal care activities compared to respondents not living with children. Net increases in consumption of unhealthy food, alcohol intake and smoking (among respondents who smoked) were observed, along with net decreases in social interaction, personal care activities and visits to the doctor and other health professionals among females residing in rural areas during the pandemic compared with pre-pandemic. It has been proposed that females will be disproportionately impacted by the pandemic due to reduced income and employment opportunities, increased carer responsibilities and worry about the virus [2]. Increased stress during the pandemic may be associated with poorer health behaviours and maladaptive coping strategies. Rural females may be at particular risk during the pandemic due to loss of usual social connections and barriers to accessing health services. 

Respondents reported mixed changes in health behaviours, with substantial proportions of respondents reporting detrimental changes in certain behaviours, while substantial proportions of respondents reporting beneficial changes in the same behaviours. This highlights the importance of asking about bidirectional changes in behaviour, as also noted by others [34]. A similar finding was reported in an Australian study that found healthy changes in physical activity, sleep, smoking and alcohol consumption were reported by 3–21% of respondents while unhealthy changes were reported by 7–49% of respondents during the pandemic [8]. Increased alcohol consumption and decreased social interaction, personal care activities and visits to doctors and other health professionals were significantly pronounced among females living with children. The increased burden of juggling work commitments with caring for children and supporting remote learning when schools and childcare centres were closed for families other than children of essential workers may have contributed to increased stress. Increased alcohol use may be an adaption strategy to cope with stress and reduced social interaction or personal care activities, and these patterns may have been more pronounced for females living with children. There is evidence that adverse health behaviour changes occur downstream of poor mental health [35]. There is also a possibility of clustering of poor health behaviours. A systematic review by Noble and colleagues reported evidence that healthy behaviours clustered together, while there was evidence of clustering between smoking and alcohol intake in approximately half of their included studies and evidence of clustering between alcohol intake, smoking, poor nutrition and physical inactivity in approximately half of included studies [36]. These findings may point to the potential for multiple health behaviour benefits when interventions address increased stress responses during the pandemic.

### 5.1. Alcohol

The increased alcohol consumption reported in our study was higher than reported by Stanton et al. (29%) [8]. Two large Australian studies reported increased alcohol use, associated with psychological distress; however, findings associated with rural residence were not assessed/reported [8,37]. Another Australian study reported that having a child caring role was strongly predictive of increased alcohol intake among females during the pandemic, and the key reason was increased stress [34]. Increased alcohol consumption, particularly at home, due to the closure of licenced venues during the pandemic is worrisome as consumption may proceed unchecked. In a British study of over sixty thousand nationally representative adults who consumed alcohol during the study period, drinking at home was associated with higher alcohol intake compared with going out, particularly for females aged 35+ [38]. Alcohol-related harm within a family context ranges from physical, emotional or sexual abuse, domestic and family violence to financial losses [39]. Parents and guardians play an important role in prevention of alcohol-related harm among young people, as evidenced in a qualitative study of young adults aged 18–20 years in the United Kingdom [40].

### 5.2. Unhealthy Food

In this study, 37% of respondents reported increased consumption of unhealthy food. Few other studies have assessed consumption of unhealthy food during the COVID-19 pandemic, particularly in Australia, and in rural areas. However, unhealthy changes to diet were reported in a recent study in Scotland and these changes were associated with low mood [41]. In another study of high school students in China, poor nutrition during the COVID-19 pandemic was higher among girls than boys, and in rural areas compared with urban areas, and was thought to be due to higher levels of stress among girls and lower socioeconomic status of rural areas [42]. 

### 5.3. Health Seeking Behaviour

Decreased visits to doctors and other health professionals during the pandemic as reported by rural female respondents were particularly pronounced among those living with children. This is a concerning finding, as important health care including women’s health checks may be being delayed. This finding is in agreement with a study from the US that reported 41% of respondents had delayed or avoided medical care during the pandemic, and this was significantly higher among females than males and those with carer responsibilities for other adults [16]. This may be due to females putting their health needs aside to prioritise care for their families, a phenomenon that has been recognised previously, for example among females with cardiovascular disease [43,44]. Although telehealth has become more available during the pandemic, it may not be suitable for all presentations [45]. Anecdotally, policies that require patients to attend face-to-face appointments alone may indirectly, but disproportionately, restrict access for women with carer responsibilities, particularly single mothers and those without social support to mind children during appointments. These policies are likely to have impacted chronic disease management and antenatal, post-natal, sexual and reproductive health services and have the potential to contribute to poor health outcomes for women and their children.

### 5.4. Psychological Distress

Mean scores for anxiety, depression and stress symptoms as per the DASS-21 scale were within the normal to mild range [31] and similar to those reported in adults in Australia during the early stages of the pandemic [8] although much lower than those reported among respondents in an Italian study [46], potentially due to higher COVID-19 infections and mortality in Italy compared with Australia. The percentages of rural females in the moderate to extremely severe range for stress, anxiety and depression (19.6%, 16.5%, 21.0% respectively) are similar to reports during the pandemic in Australia (18.1%, 13.5%, 26.5% respectively) [8]. Interestingly, females living with children did not have significantly higher stress scores than females living without children, but each group exhibited significantly higher stress scores than the normative scores for Australian females [31]. The net decrease in social interaction and personal care activities, particularly among those living with children, may be important to consider in the context of maintenance of mental health and wellbeing [47].

## 6. Implications

Females residing in rural areas appear to be experiencing potentially detrimental changes to health behaviours, particularly those living with children. The association between health behaviours and the stress of the pandemic is important to understand as unhealthy behaviour changes can increase the risk of development of chronic disease or exacerbation of pre-existing health conditions, which can be compounded if people are not engaging with healthcare providers at their usual level. Certain groups of people, such as females, those with carer responsibilities and rural people may require additional support during the pandemic to optimise their health and wellbeing and reduce their stress and risk of chronic ill health. This study shows that gender responsive, if not specific, policies are required to address the implications of the COVID-19 pandemic [48]. There have been calls for the establishment of a Taskforce on Gender, Mental Health and Disaster to address similar concerns [49] and for increased representation of women in policy and decision-making roles during the pandemic [48]. 

## 7. Limitations

The results of this study are based on a convenience sample of employees and volunteers recruited online, and the sub-sample of females and residents of rural areas are presented in this paper. The convenience sample from which this sub-sample was drawn was over-represented by females (80%) and people residing in rural areas (78%) and is therefore non-representative of the Australian population (51% females, 29% rural residents ) [50]. The median age of the convenience sample was 43 years, older than the median age of the Australian population (38 years) [50]. The limitations of this convenience sample are somewhat tempered by inclusion of the sub-sample of rural females or a homogenous convenience sample [51]. The reliability and validity of the survey tool will need to be assessed in future studies. There is the potential for social desirability bias to have influenced results, particularly among respondents within the authors’ networks. Online surveys may be less accessible to people with lower literacy or computer literacy and lower socioeconomic status. The questions related to health behaviour were single questions and did not capture the degree of health behaviour (for example amount of exercise or alcohol intake) either prior to or during the pandemic. Self-reported presence of chronic physical or mental health conditions and health behaviours may underestimate poor health and poor health behaviour, as reviewed by Newell and colleagues [52]. Observational studies such as this are unable to assess causality. 

## 8. Future Research

Further research is required to better understand changes to health behaviours and ways to support people, particularly people in rural areas, women and families, during crises at a systems level. Workplaces play a crucial role in employee health and wellbeing during and after pandemics, as they are pivotal to retaining connection and sense of purpose. It is important that employers ensure that policies support female employees and employees with carer responsibilities during stressful periods. The inclusion of questions related to sleep patterns and screen time would be valuable in future studies. Longitudinal studies or repeat cross sectional studies will be important to better understand the longer-term associations between the restrictions on everyday life that have been necessary to combat the COVID-19 pandemic, stress and health behaviours. We plan to repeat this study after restrictions to everyday life have lifted. 

## 9. Conclusions

The COVID-19 pandemic and restrictions associated with the public health response have been associated with potentially adverse changes in a range of health behaviours, social interaction and health seeking behaviours. Women living with children may be at heightened risk of unhealthy changes and may require additional support during the pandemic. Gender responsive policies are required to avoid worsening health and social inequities associated with the COVID-19 pandemic.

## Figures and Tables

**Figure 1 ijerph-18-00722-f001:**
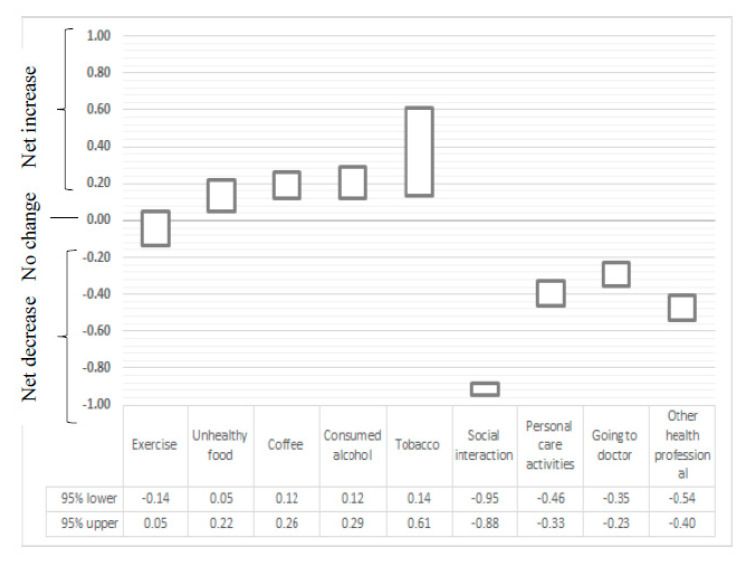
Average net change in behaviours reported by respondents (95% confidence interval) Decreased behaviours were coded as –1, unchanged as 0 and increased behaviours as +1.

**Table 1 ijerph-18-00722-t001:** Demographic characteristics and DASS-21 scores of female respondents residing in rural areas.

	Total *n* (%), Unless Specified
Respondents *n*	339
Diagnosed with COVID-19 *n*	0
Living arrangements	
• Lives alone	49 (14.5)
• Lives with partner (with or without children) *	221 (65.2)
• Lives with pre-school aged children *	43 (12.7)
• Lives with school aged children *	110 (32.4)
Pre-existing chronic disease	
• yes	69 (20.5)
• no	261 (77.4)
• don’t know	4 (1.2)
• prefer not to say	3 (0.9)
Pre-existing mental health condition	
• yes	80 (23.7)
• no	250 (74.0)
• don’t know	3 (0.9)
• prefer not to say	5 (1.5)
Tobacco smoking	
• yes	32 (9.8)
• no	295 (90.2)
Age, mean ± SD	44.6 ± 12.6 (range 18–77)
Primary work	
• Full time	163 (48.2)
• Part time	143 (42.3)
• Casual	22 (6.5)
• Not applicable	1 (0.3)
• Other	9 (2.7)
Impact of pandemic on primary work	
• Temporarily stood down	10 (3.1)
• Role terminated	5 (1.5)
• Hours reduced	24 (7.4)
• Hours and role unchanged	212 (65.2)
• Other	74 (22.8)
Role	
• In paid work	331 (97.9)
• Volunteer	7 (2.1)
DASS stress score mean ± SD	5.9 ± 4.5
• DASS stress normal to mild	254 (80.4)
• DASS stress moderate to extremely severe	62 (19.6)
DASS anxiety score mean ± SD	2.4 ± 3.3
• DASS anxiety normal to mild	263 (83.5)
• DASS anxiety moderate to extremely severe	52 (16.5)
DASS depression score mean ± SD	3.9 ± 4.3
• DASS depression normal to mild	248 (79.0)
• DASS depression moderate to extremely severe	66 (21.0)
DASS-21 total score mean ± SD	12.2 ± 11.0

* Multiple responses possible.

**Table 2 ijerph-18-00722-t002:** Changes to behaviour among females residing in rural areas with or without children.

	Total *n* (%)	Lives without Children *n* (%)	Lives with Children *n* (%)	*p*
**Exercise**				
Decreased	119 (37.1)	69 (36.3)	50 (38.2)	0.736
Unchanged or increased	202 (62.9)	121 (63.7)	81 (61.8)	
**Consumption of unhealthy food**				
Increased	113 (36.7)	63 (35.2)	50 (38.8)	0.522
Unchanged or decreased	195 (63.3)	116 (64.8)	79 (61.2)	
**Consumption of alcohol**				
Increased	104 (38.8)	48 (30.4)	56 (50.9)	0.001
Unchanged or decreased	164 (61.2)	110 (69.6)	54 (49.1)	
**Tobacco smoking**				
Increased	15 (46.9)	7 (38.9)	8 (57.1)	0.305
Unchanged or decreased	17 (53.1)	11 (61.1)	6 (42.9)	
**Consumption of coffee**				
Increased	70 (28.1)	37 (25.9)	33 (31.1)	0.361
Unchanged or decreased	179 (71.9)	106 (74.1)	73 (68.9)	
**Social interaction**				
Decreased	299 (91.7)	171 (89.1)	128 (95.5)	0.037
Unchanged or increased	27 (8.3)	21 (10.9)	6 (4.5)	
**Personal care activities**				
Decreased	152 (47.2)	75 (39.5)	77 (58.3)	0.001
Unchanged or increased	170 (52.8)	115 (60.5)	55 (41.7	
**Visiting the doctor**				
Decreased	106 (34.6)	50 (28.1)	56 (43.8)	0.005
Unchanged or increased	200 (65.4)	128 (71.9)	72 (56.3)	
**Visiting other health professional**				
Decreased	145 (51.8)	74 (45.7)	71 (60.2)	0.017
Unchanged or increased	135 (48.2)	88 (54.3)	47 (39.8)	

**Table 3 ijerph-18-00722-t003:** Associations between rural females living with or without children and changes to behaviour.

	Logistic Regression-Odds Ratio (95% CI)		
Dependent Variable	Not Living with Children	Living with Children	*p*
Increased alcohol consumption	1	2.373 (1.359, 4.145)	0.002
Decreased social interaction	1	2.883 (1.035, 8.033)	0.043
Decreased personal care activities	1	2.342 (1.384, 3.963)	0.002
Decreased visits to doctor	1	1.850 (1.076, 3.183)	0.026
Decreased visits to other health professionals	1	1.932 (1.118, 3.337)	0.018

Separate logistic regressions were completed for each dependent variable (increased alcohol consumption, decreased social interaction, decreased personal care activities, decreased visits to doctors or other health professionals). Included in adjusted model: n = 339. Reference: no change or decreased alcohol consumption, no change or increased social interaction, no change or increased personal care activities, no change or increased visits to doctor or other health professionals. Adjusted for independent variables of living with children, age, DASS-21 stress score, DASS-21 anxiety score, DASS-21 depression score and self-reported chronic disease.

## Data Availability

The data presented in this study are available on request from the corresponding author. The data are not publicly available due to privacy issues.

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
