# Peer review of "Detrimental Health Behaviour Changes among Females Living in Rural Areas during the COVID-19 Pandemic"

_ijerph, 2021, doi:10.3390/ijerph18020722_

Round 1
Reviewer 1 Report
This is a generally well-written article that provides a strong rationale of the need for this study and draws reasonable conclusions based on study findings. Most of my suggested revisions will focus on the description of the methods and reporting of the results which are the weakest aspects of the paper. However, I believe these limitations can be addressed by the authors. I also think it is an important paper to publish. The following are my suggestions to improve the paper:
- The primary title reflects the content of the manuscript; however, I would remove the subtitle as the authors did not formally compare women to men or rural women to urban women.
- The literature could have been strengthened by referencing a theoretical framework that attempts to explain behavior change (and that would incorporate some of the variables that they ultimately examine in their analysis).
- I would suggest more specific study aims that serve as a guide for your analysis and reporting of results. At present, the analysis and results sections appear somewhat unsystematic (ie., like you were testing many associations rather than testing particular hypotheses). Based on your results it seems like living with or without children is your primary exposure of interest.
- More detail is required regarding the electronic survey and the survey link ie., “authors personal and professional networks via email and social media with subsequent snowball recruitment”. I don’t have a good sense of who the potential participants are. Is there any information on response rate? Can the authors provide any evidence of the validity of this approach. Have others used these methods?
- More detail is required on the study variables: the actual questions, the form in which they were used (e.g. categorical, continuous) and why and evidence of validity/reliability. Ideally, there would be a proper measures section with independent and dependent variables clearly identified, as well as potential confounders/covariates. It is difficult to evaluate the quality of measures with so little detail provided. The analysis section should be separated from the measures section, with more detail present and with a logical flow from descriptive to multivariable. Ideally, this analysis plan would be more explicitly linked with research aims/questions.
- For each variable in Table 1, all categories should be included (ie., live alone: yes (n%) and no(n%), so each adds up to 100%. The title says demographic but I don’t think the DASS scores would be considered that.
- Some of the results according to living with children or not get lost in the text; it would be better to report them in a table similar to Table 2.
- Table 3 is unclear. I am assuming you did a series of separate logistic regression analyses for each of the dependent variables listed with living with children/not as the primary exposure (and adjusting for various covariates. The interpretation in text is unclear as to which variable is the independent/dependent variable. See suggest below (need to add in covariates, 95% CIs, proper title, and that separate logistic regression analyses were done – all tables need to stand alone).
|
|
Increased alcohol |
Decreased social interaction |
Decreased personal care activities |
Decreased doctor visits |
Decreased other health professional visits |
|
Not living with children |
1.00 |
1.00 |
1.00 |
1.00 |
1.00 |
|
Living with children |
2.37 |
2.88 |
2.34 |
1.85 |
1.93 |
I think the manuscript could be improved significantly if the authors presented their aims as more specific research questions (with a primary exposure in mind), which then can be more clearly linked to their methods, analyses and reporting of results.
Reviewer 2 Report
Detrimental health behavior changes among females living in rural areas during the Covid-19 pandemic; Same storm but different boats
Comments to the authors
Thank you for the opportunity to review this manuscript. The work presented here is important and compelling. Overall, the paper is well written with a clear objective which seems to guide the authors throughout the different sections. However, there are some concerns about the manuscript in its current form that I would like to see addressed. Here are some comments that I hope will be helpful to the authors and future readers of the paper if published.
Title: In the title, there is this statement, ‘same storm but different boats.’ As a reader, I am not sure how this applies to the research. It might be good for the authors to see how they can bring in this interesting quote into the paper so that it does not seem misplaced.
Abstract – could also benefit from specific results including the 95% CI values
Introduction
The introduction is generally well written and establishes some bits and pieces of the background and the justification of the study, thereby creating a great start for the reader. May require some editing but overall, it is good.
Materials and methods
The methods section also requires some beefing up to enable the reader to understand how the study was conducted and for replicability. In paragraph starting from line 134, the authors talk about DASS-21 sub-scales data collection tool. As the reader, I feel that I am unable to really understand clearly what was measured here. It will be great to add more information on the tools used. Also, additional details on the statistical analysis will be great. There is the mention of independent t-test, but it is unclear to me the kind of data that were collected and analyzed using this method. I am imagining that the results in figure 1 might be the once based on this analysis. For clarity, I suggest that you be upfront in the methods on the kind of data collected and the analysis done so that the reader do not struggle to find some of this information.
Results
Table 1: The results in this take is a little confusing as presented as the total for the different categories do not add to 100%. E.g., pre-existing mental health conditions # 80(24.2%). There’s not percentage for those without pre-existing mental health. Same applies to other demographic characteristics.
In line 154-160, a further analysis is done but I do not know what analysis this is. I am not sure if the results summarized in this paragraph are presented in a table or a figure anywhere else in the paper.
Figure 1: Not sure what kind of data were collected and what analysis was done to generate the figure. Additional information in the methods will suffice.
Lines 176-183 – presenting/summarizing data in Table 3 may benefit from having the OR, 95%CI within the text so that the reader do not have to shift between the text and the table while reading. A further interpretation of the ORs is also necessary in this section so that the reader can make more sense of the results. I am also being curious here and just wondering what happens to the ORs if all the other variables are included in the logistic model? Just something to think through
Discussion & conclusion
This section is well written as it clarifies some of the concerns and the conclusions are drawn from the research findings. However, a clear presentation of the results will strengthen the discussion even more as it will allow the reader to engage more with the findings while seeing how the findings relate to other pieces of work in the same topic.
Limitations
It is good that the authors recognize some of the study limitations – such as the sampling method used, the reduction of the sample size to only include females in rural areas. The question that comes to mind is that are their ways that the authors ensure reliability and validity of their findings? Acknowledging this is good but not enough. Issues of social desirability could also impact the responses given that the samples were based on the social networks of the researchers and it is worth acknowledging that as well.
Reviewer 3 Report
This is a report of a survey examining the impact of the COVID-19 pandemic on important health behaviours among rural women in Australia and in particular on those living with and without children. The authors report a number of concerning findings with respect to this group that have important implications for future public health responses. The manuscript is succinct, clear and well written. The analysis is sound and the discussion covers important ground and makes some clear recommendations. This is a good study that contributes important information to the study of the impacts of the COVID-19 pandemic. My main suggestion for improvement relates to being more explicit about the aim with respect to the focus on the impact of living with children. I have made some specific suggestions with respect to this below and added a number of other suggestions for improvement.
Major comments
- Introduction & Aim – the authors focus the analysis on comparisons between women with and without children, yet the overall title and aim is focused on rural women generally. It strikes me that both aims are relevant and could be acknowledged. There is some good general justification for the split between children vs not in the intro but could the authors also add an explicit justification for this in the final para of the intro (L90-95) and to the aim in the abstract and at the end of the intro (L90-95). Did the authors have any specific hypotheses about this?
- Methods: The authors adjusted the logistic regression for “chronic health conditions”. What was included in this? Could this be clarified in the methods please.
- Methods – Analysis: Why were changes in eating not included in the logistic regression? I assume it is because these comparisons were not significant in unadjusted tests? Could the authors clarify the analyitic approach in the methods please.? Ie sig covariates were included in a logistic regression controlling for XXX to further examine the association of behaviour changes with child carer status
- Discussion – the first paragraph focuses on the overall results but not the findings with respect to living with and without children. It would be useful to have this here before the full discussion that follows. This is important as the analysis indicated that most negative behaviour changes were associated with living with children.
- Discussion: Alcohol L222-L236. Could the authors comment on the impact of living with children here?
- Please comment on the limitations of self-report with respect to measuring of these behaviours. Could the authors also comment on how the convenience sample generally compares to other representative samples. Do the authors have any demographic info to compare? Age, education, other things?
Minor comments
- Methods: Survey tool. Could the authors provide some more detail about the questions used to assess the behaviours in the current study. Were they adapted from anywhere else?
- How did the authors measure “pre-existing mental health conditions” and also chronic health conditions? Could they add this to the methods please?
- Table 1. The formatting is very hard to read. I assume the editorial team could work with the authors to reformat and match journal style but at the least the n(%) should appear in the heading of Column 2 as it seems to apply to all rows with the exception of DASS scores where mean (SD) and N(%) formats are used. Please let justify too for clarity (and this applies to some other tables too)
Round 2
Reviewer 1 Report
The authors have satisfactorily addressed all of my suggested revisions.